# Better Performance of the Modified CERES-Wheat Model in Simulating Evapotranspiration and Wheat Growth under Water Stress Conditions

Yingnan Wei [1,2], Han Ru [1], Xiaolan Leng [1], Zhijian He [1], Olusola O. Ayantobo [3], Tehseen Javed [4] and Ning Yao [1,*]

1   College of Water Resources and Architectural Engineering, Northwest Agriculture and Forestry University, Xianyang 712100, China
2   Key Laboratory for Agricultural Soil and Water Engineering in Arid Area of Ministry of Education, Northwest A&F University, Xianyang 712100, China
3   State Key Laboratory of Hydroscience and Engineering, Department of Hydraulic Engineering, Tsinghua University, Beijing 100086, China
4   Department of Environmental Sciences, Kohat University of Science and Technology, Kohat 26000, Pakistan
*   Correspondence: yaoning@nwafu.edu.cn

**Abstract:** Crop models are important for understanding and regulating agroecosystems. Although the CERES-Wheat model is an important tool for winter wheat research, it has some limitations under water stress conditions. To narrow the gap, this study aimed to improve the performance of the CERES-Wheat model under water stress in arid and semi-arid regions based on the winter wheat experimental data from 2012 to 2014. The Priestley–Taylor (PT) and FAO56 Penman–Monteith (PM) equations were used to calculate the reference crop evapotranspiration and further modified the crop coefficient of the CERES wheat model to improve the simulation accuracy of crop yield and evapotranspiration under water stress conditions. The results showed that: water stress before jointing seriously affected the accuracy of the CERES-Wheat model in simulating biomass and grain yield, so it was necessary to improve the original model. In the original and improved models, the accuracy of the PM equation was lower than that of PT. In addition, the simulation accuracy of the improved model was higher than that of the original model (the average *RMAE* and *RRMSE* are less than 30%). In general, among the four scenarios, the PT equation for calculating crop reference evapotranspiration and crop coefficient had the best performance. Water stress occurred at the heading and grain filling stages, and the simulated biomass was in good agreement with the observed results, which better simulated the soil water content under water stress at the later growth stages. Therefore, the change in water stress response function had positive effects on winter wheat growth under simulated water stress conditions. This study provided a reference for applying the CERES-Wheat model in arid and semi-arid areas.

**Keywords:** evapotranspiration; dynamic crop coefficient; water stress; CERES-Wheat; winter wheat

## 1. Introduction

The global wheat cultivation area is 214 million hectares, with a total output of 762.06 million tons, ranking second among cereal crops [1]. Winter wheat (*Triticum aestivum* L.) is one of the main staple crops in the arid and semi-arid regions of northwest China. However, winter wheat production is constrained by increasingly tight water supplies, which causes farmers to consider limited irrigation as an alternative to full irrigation practices. Therefore, many studies are exploring the benefits of limited or deficit irrigation on winter wheat.

Due to the time-consuming, expensive, and lack of generality for traditional field methods, crop models have become valuable tools for predicting crop growth, development, and yields based on meteorological and environmental conditions and agricultural

management [2,3]. The models greatly simplify and shorten the research processes of agricultural systems and are, therefore, suitable alternatives for field experiments, saving time and economic costs. Additionally, crop models have been widely employed to optimize farming management practices and provide technical guidance for farmers and decision-makers [4,5]. In recent years, crop models have been increasingly applied for agricultural decision-making and management at local, regional, and global scales [6–8].

Major wheat simulation models include the CERES-Wheat in the Decision Support System for Agrotechnology Transfer (DSSAT) crop system model [9], SWHEAT [10], SUCROS2 [11], AFRCWHEAT2 [12], STICS [13], Sirius [14], etc. DSSAT was developed by the United States Department of Agriculture for agricultural experiment analysis, yield prediction, agricultural production risk assessment, etc [3,15,16]. It has a simple interface and a wide application range. The DSSAT-CERES-Wheat model is a sub-module of the DSSAT model, which mainly simulates the growth process of winter wheat. The DSSAT-CERES-Wheat requires inputs of irrigation schemes and meteorological and soil information, which can comprehensively consider the impact of soil, weather, management, and other factors. The output data mainly include winter wheat phenology, soil moisture, leaf area index, and yield. The CERES-Wheat model has been applied widely in irrigation scheduling [17,18], fertilizer management [19,20], plant breeding [21], and influences of climate change on wheat production around the world [22–24]. For example, He et al. [25] applied the CERES-Wheat model to simulate spring wheat growth and development under different irrigation practices in Minqin, Gansu, China and selected an optimal irrigation schedule. Yao et al. [26] used the CERES-Wheat model to study the effects of drought change on leaf area index (LAI), biomass, and yield of wheat in arid and semi-arid northern regions, Qinghai-Tibet Plateau, Huang-Huai-Hai Plain, and northeast Plain from 1961 to 2018. The CERES-Wheat model can also efficiently simulate soil water dynamics, phenology, biomass, yield and nitrogen in Pakistan [20,27]. Wajid et al. [28] found that the simulated crop phenology, LAI, total dry matter, and yield using the CERES-Wheat model had a high degree of consistency with the measured data. Thus, the CERES-Wheat model could be used as a decision support system tool for crop improvement in a fragile environment.

Moreover, the CERES-Wheat model has been applied under sufficient water supply conditions [29–31] and water stress conditions [32,33]. However, it had some limitations under water stress conditions. For instance, in New Zealand, grain yield simulation accuracy was low during drought conditions in various growing stage treatments, with a root mean square error (*RMSE*) of over 3000 kg hm$^{-2}$ [34]. An *RMSE* of 897 kg ha$^{-1}$ has been reported when comparing the simulated grain yields with the consensus values across 141 wheat-growth sites in China [35]. Tian et al. [36] observed that the percentage error in grain yields simulated at 36 stations in China ranged between 11.6% and 33.6%, depending on the calibration methods. Yao et al. [37] also conducted field experiments on winter wheat growth under a movable rain shelter with different water stress treatments in Yangling, Shaanxi, China. They found that the CERES-Wheat model had low accuracy with discrepancies in phenology dates of winter wheat. There were relatively large simulation errors between observed and simulated yield when water stress occurred at the wintering and returning green stages. Yan et al. [32] found that the CERES-Wheat model simulated yield, maximum leaf area index, and above-ground biomass poorly under water stress conditions compared to adequate water.

The poor performance of CERES-Wheat under water stress conditions brought the necessity of modifying it to improve its performance since water stress has severe impacts on agricultural production, especially in arid and semi-arid regions of northwest China [32,33,38]. The simulation errors could result from the low accuracy of simulated potential crop evapotranspiration ($E_0$). The CERES-wheat uses a static crop coefficient ($K_{CS}$) multiplied by reference crop evapotranspiration ($ET_0$) to evaluate $E_0$, which only works well for non-stressed water conditions. Dejonge et al. [39] found that CERES-Maize overestimated evapotranspiration (*ET*) under deficit irrigation. Soldevilla-Martinez et al. [40] pointed out that the DSSAT model was not accurate in simulating daily *ET* variations. Kheir et al. [41]

found that the water use efficiency of simulation-based evapotranspiration increased with the increase of irrigation with a slight increase; under the condition of low water value is low, the simulation effect is poor. Therefore, the simulated accuracy of *ET* must be improved under water stress conditions, improving yields' simulation and crop model performance.

So far, the poor accuracy of *ET* and yield simulation of CERES-Wheat under water stress remains unresolved. To address this, we assume that using the dynamic crop coefficient could improve the performance of the CERES-Wheat. Therefore, the objective of this work is to evaluate total soil water dynamics accuracy and winter wheat yield-related indices simulated by the non-modified and modified CERES-Wheat model using experimental data from various sites. This research would provide insights and guidance for the broader application of CERES-Wheat in arid and semi-arid areas.

## 2. Materials and Methods

### 2.1. Field Experiment

Field data of winter wheat from three experimental field sites provided by Yao et al. [37] and DSSAT v4.7 databases [42] were employed for modifying and validating the CERES-Wheat model. The sites and aims of different experiments are: (i) winter wheat experiments in the irrigation test station of Northwest A&F University, Yangling, China (34°17′ N, 108°04′ E) during 2012–2014 were used for the modifications of $K_{CS}$, The wheat variety 'XiaoYan 22' was used, and the sowing dates were 15 October 2012 and 2013. Strip planting was adopted, and samples were taken every two weeks before the jointing stage and once a week after the jointing stage. The soil parameter data of three experimental field sites are shown in Table S1. (ii) Winter wheat experiments located in Kansas State, USA (37.18° N, 99.75° W) and Rothamsted, England (52.50° N, 0.50° W) were employed for validating the modified CERES-Wheat model. Details of the experimental designs are presented in Table 1. The *ET* was determined by field observations of soil water content within depths of 0–100 cm using water balance.

**Table 1.** The experiment information of irrigation for the three experiments used for modifying and validating the CERES-Wheat model.

| Cultivar | Sowing date | Treatment | Irrigation Amount (mm) | | | | | Harvest Date | Reference | Aim |
|---|---|---|---|---|---|---|---|---|---|---|
| XiaoYan 22 (China) | 15/October/2012 15/October/2013 | Date | 12/15 | 3/15 | 4/15 | 5/1 | 5/15 | 2/June/2013 7/June/2014 | Yao et al. [37] | Modification |
| | | I1D1 | - | - | 40 | 40 | 40 | | | |
| | | I1D2 | 40 | - | - | 40 | 40 | | | |
| | | I1D3 | 40 | 40 | - | - | 40 | | | |
| | | I1D4 | 40 | 40 | 40 | - | - | | | |
| | | I2D1 | - | - | 80 | 80 | 80 | | | |
| | | I2D2 | 80 | - | - | 80 | 80 | | | |
| | | I2D3 | 80 | 80 | - | - | 80 | | | |
| | | I2D4 | 80 | 80 | 80 | - | - | | | |
| | | CK | 80 | 80 | 80 | 80 | 80 | | | |
| NEWTON (USA) | 16/October/1981 | Date | 4/6 | 4/20 | 4/27 | | | 23/6/1982 | DSSAT databases | Calibration |
| | | T1 | - | - | - | - | - | | | |
| | | T2 | - | - | - | - | - | | | |
| | | T3 | - | - | - | - | - | | | |
| | | T4 | 65 | 78 | 70 | - | - | | | |
| | | T5 | 65 | 78 | 70 | - | - | | | |
| | | T6 | 65 | 78 | 70 | - | - | | | |
| MARIS FUNDIN (England) | 6/November/1974 | T1–T8 | - | - | - | - | - | 1/8/1975 | DSSAT databases | Calibration |

Daily solar radiation (*SRAD*, MJ m$^{-2}$), maximum temperature (*TMAX*, °C), minimum temperature (*TMIN*, °C), and rainfall (*RAIN*, mm) are shown in Figure 1. Rainfall during the growing winter wheat season in Kansas State and Rothamsted are 600.3 mm and 558.1 mm, respectively.

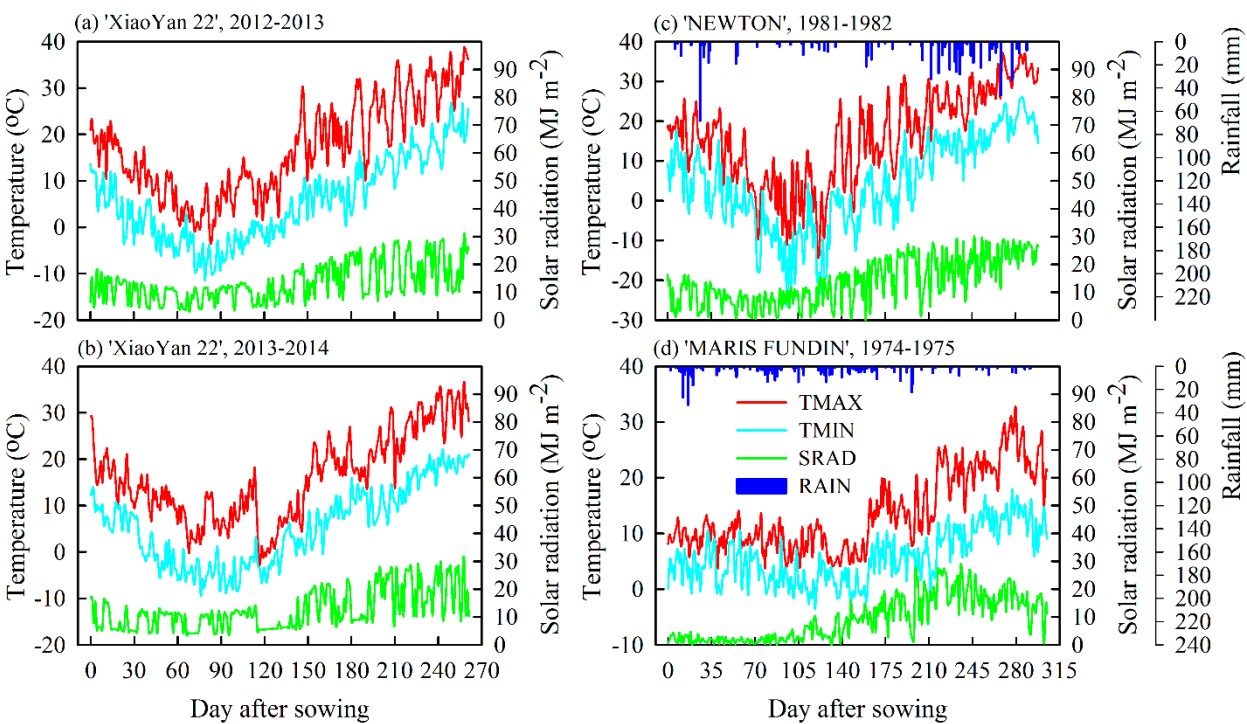

**Figure 1.** Daily solar radiation, maximum temperature, minimum temperature, and precipitation during winter wheat growing seasons in three experimental field sites. The *SRAD*, *TMAX*, *TMIN*, and *RAIN* represent daily solar radiation, maximum temperature, minimum temperature, and rainfall, respectively.

### 2.2. General Description of the CERES-Wheat Model

The CERES-Wheat model is embedded in the CSM platform (DSSAT v4.7), is part of the Decision Support System for Agrotechnology Transfer, and is capable of simulating soil carbon and nitrogen, water balances, growth and development, and yields of wheat at a daily step [3]. It requires meteorological, soil, crop genetic coefficient, and management data. Meteorological data include daily maximum temperature, daily minimum temperature, precipitation, and solar radiation. Soil data include soil type, physical soil properties of each layer, and initial physical and chemical properties of soil. Crop genetic parameters include ecotype, species type, and variety type. Field management data include crop variety, sowing time, sowing method, irrigation management, fertilization management, and harvest time. In addition, DSSAT-CERES-Wheat usually simulates canopy solar radiation uptake and photosynthesis, root nutrient uptake and activity, assimilate partitioning, water uptake and transpiration, growth and respiration, leaf area expansion, organogenesis and senescence, and field management in days [18,43–46]. It has been proven that DSSAT-CERES-Wheat can optimize irrigation regimes and increase the economy of agricultural production [3].

The basic principle for determining water stress is to compare potential transpiration ($EP_0$) and potential root water uptake (*TRWUP*) [47]. Under well-watered conditions, *TRWUP* is higher than $EP_0$. As soil water content decreases with root water uptake and surface evaporation, *TRWUP* also decreases. At a particular stage, a threshold is reached where the first water stress factor (*TURFAC*, Equation (1)) is activated, which mainly affects wheat expansive growth. When $EP_0$ equals or exceeds the *TRWUP*, a second stress factor (*SWFAC*, Equation (2)) is activated [48]. *SWFAC* mainly affects growth and biomass-related processes.

$$TURFAC = TRWUP / (RWUEP_1 \times EP_0) \tag{1}$$

$$SWFAC = TRWUP / EP_0 \tag{2}$$

where $RWUEP_1$ is a species-specific parameter that is set to 1.5. The *TURFAC* and *SWFAC* values are set to 1.0 under non-water-stress conditions. $EP_0$ is estimated as follows:

$$EP_0 = E_0 \times (1 - \exp(-KEP \times LAI)) \tag{3}$$

where *KEP* (default value of 0.685) is defined as an energy extinction coefficient of the canopy for total solar irradiance; *LAI* is leaf area index; $E_0$ is potential crop evapotranspiration, which is calculated as the product of static crop coefficient and reference evapotranspiration, mm.

$$E_0 = K_{CS} \times ET_0 \tag{4}$$

$ET_0$ can be calculated either with the Priestley–Taylor [49] or with the FAO56 Penman–Monteith equations [50].

The Priestley–Taylor (PT) equation for computing $ET_0$ is written as:

$$ET_0 = EEQ = 2.04 \times 10^{-4} \cdot R_s - 1.83 \times 10^{-4} \times A_{LBEDO} \cdot (0.6TMAX + 0.4TMIN + 29) \tag{5}$$

where *EEQ* is the daily equilibrium *ET,* mm d$^{-1}$; $A_{LBEDO}$ is the crop albedo; *TMAX* and *TMIN* are maximum and minimum temperatures, respectively, °C; $R_s$ is total solar radiation, MJ m$^{-2}$ d$^{-1}$.

The FAO56 Penman-Monteith (PM) method was adopted to calculate daily $ET_0$ (mm day$^{-1}$), which was proposed by Allen et al. [50]:

$$ET_0 = \frac{0.408\Delta(R_n - G) + \gamma\left(\frac{900}{T+273}\right)U_2(e_s - e_a)}{\Delta + \gamma(1 + 0.34U_2)} \tag{6}$$

where $\Delta$ is the slope of saturated vapor pressure, KPa °C$^{-1}$; $T$ is the daily average air temperature at 2 m, $T = 0.5 (T_{\max} + T_{\min})$, °C; $\gamma$ is psychrometric constant, KPa °C$^{-1}$; $R_n$ is net radiation, MJ m$^{-2}$ day$^{-1}$; $G$ is downward ground heat flux, MJ m$^{-2}$ day$^{-1}$, $G = 0$ for daily timescale; $e_{sat}$ and $e_a$ are saturated and actual vapor pressures, KPa.

For the PT method, the $K_{CS}$ value is a function of maximum temperature (*TMAX*).

$$K_{CS} = \begin{cases} 0.01 \times EXP(0.18 \times (TMAX + 20)) & TMAX < 5\,°C \\ 1.1 & TMAX \in [5, 35]\,°C \\ 0.05 \times (TMAX - 35) + 1.1 & TMAX > 35\,°C \end{cases} \tag{7}$$

For the PM method, the $K_{CS}$ value is a function of the leaf area index (*LAI*):

$$K_{CS} = 1.0 + (EORATIO - 1.0) \times LAI/6.0 \tag{8}$$

where *EORATIO* is defined as the maximum $K_{CS}$ at *LAI* = 6.0 [51,52]. For the wheat, the parameter is hardcoded to *EORATIO* = 1.0. This fixes $K_{CS}$ at 1.0 for the entire simulation, making it static and limiting mid-season crop coefficient options for wheat, which has recommended mid-season $K_C$ values of 1.15 and above Allen et al. [50].

### 2.3. Modification of $K_{CS}$ in CERES-Wheat

To modify the CERES-Wheat under water stress conditions, it is necessary to link the factors governing the interception of solar radiation (such as *LAI*) with those governing *ET* demands to provide a more mechanistic representation of the *ET*. Kang et al. [53] directly compared crop coefficient ($K_C$) and *LAI* and provided a function to compute $K_C$ and *LAI* (Equation (9)). An exponential decay function for dynamic crop coefficient $K_{CD}$ is proposed by Dejonge et al. [39] to replace the original crop coefficient $K_{CS}$ (Equation (10)):

$$KC_{Kang} = K_{Cmin} + 0.8466LAI/(0.7887 + LAI) \tag{9}$$

$$KC_D = K_{Cmin} + (K_{Cmax} - K_{Cmin})(1 - \exp(-SK_C \times LAI)) \tag{10}$$

where $K_{Cmin}$ and $K_{Cmax}$ are minimum and maximum crop coefficients, which are 0.4 and 1.15, respectively. $SK_c$ is a shaping parameter (=0.8).

### 2.4. Model Calibration and Verification

The generalized likelihood uncertainty estimation (GLUE) was used to estimate genetic parameter values. The GLUE output variables include the unit grain weight (*HWUM*, mg), above-ground biomass (*CWAM*, kg hm$^{-2}$), grain yield (*HWAM*, kg hm$^{-2}$), and important phenology dates (e.g., anthesis date, *ADAP*, and maturity date, *MDAP*). There are two rounds of model runs in the current DSSAT-GLUE package. The genetic coefficients related to crop phenology (*ADAP* and *MDAP*) and coefficients related to crop growth (*HWUM*, *CWAM*, and *HWAM*) were estimated in the first and second rounds, respectively. The 20,000 model runs were recommended in each GLUE round to ensure that the genetic coefficients were each estimated accurately and the posterior distributions were reliable He [54]. For the cultivars 'XiaoYan 22′, the parameters *G2* (41.7) and *PHINT* (130) were determined according to the observed values of *HWUM* and leaf number (nine leaves). Yan et al. [32] and Li et al. [55] validated wheat variety parameters using GLUE and indicated good performance. Other parameters were estimated using the observation data of *CK* treatment collected in the 2012–2013 and 2013–2014 growing seasons. Winter wheat cultivars 'NEWTON' and 'MARIS FUNDIN' are provided in the DSSAT database (Table 2).

**Table 2.** Genetic coefficients of different winter wheat cultivars in the CERES-Wheat model.

| Parameter | Definition | Value | | |
|---|---|---|---|---|
| | | XiaoYan 22 | NEWTON | MARIS FUNDIN |
| P1V | Days at the optimum vernalizing temperature required to complete vernalization (d) | 41 | 45 | 30 |
| P1D | Photoperiod response (% reduction in rate/10 h drop in pp, %) | 93 | 75 | 83 |
| P5 | Grain filling (excluding lag) phase duration (°C d) | 621 | 500 | 515 |
| G1 | Kernel number per unit canopy weight at anthesis (# g$^{-1}$) | 22 | 25 | 15 |
| G2 | Standard kernel size under optimum conditions (mg) | 41.7 | 30.0 | 44.0 |
| G3 | Standard, non-stressed mature tiller wt (incl grain) (g) | 1.0 | 2.0 | 3.2 |
| PHINT | The interval between successive leaf tip appearances (°C d) | 130 | 95 | 100 |

### 2.5. Statistical Method

Relative mean absolute error (*RMAE*) and relative root mean square error (*RRMSE*) are used to evaluate the performance of the original and modified CERES-Wheat models by comparing the simulated and observed *ADAP*, *MDAP*, *HWUM*, *CWAM*, and *HWAM*. Both tools are dimensionless in comparing different output variables [56]:

$$RMAE = \frac{1}{n}\sum_{i=1}^{n}\frac{|S_i - O_i|}{|O_i|} \times 100\% \tag{11}$$

$$RRMSE = \frac{\sqrt{\frac{1}{n}\sum_{i=1}^{n}(S_i - O_i)^2}}{\overline{O}} \times 100\% \tag{12}$$

where $O_i$ and $S_i$ are the *i*-th observed and simulated values, respectively; $\overline{O}$ is mean observed value, and *n* is the total number of values.

### 3. Results

#### 3.1. Calibration and Validation of the Original CERES-Wheat Model

The simulation accuracy of grain yield (*HWAM*) was high, the *RMAE* value was 2.8%, and the *RRMSE* value was 3.0% (Figure 2b). In contrast, the simulated *ET* value was low, with *RMAE* and *RRMSE* > 20% (Figure 2c). For the validation process, *RMAE* and *RRMSE* at flowering and mature stages are less than 3.0% (Figure 2a). However, the

simulated anthesis and maturity dates were the same for different treatments each year, while observed dates differed because of various water stress scenarios. The simulation accuracy of grain yield and *ET* was relatively poor, especially for the treatments with water stress at returning green stage heading. The *HWAM* was underestimated seriously, with *RMAE* and *RRMSE* values of *HWAM* and *ET* close to 20%, and even *the RRMSE* exceeded 30% for *HWAM* (Figure 2b,c). This indicated that water stress before jointing could greatly influence the model simulation performance of biomass and grain yield. Therefore, it is necessary to improve the original model.

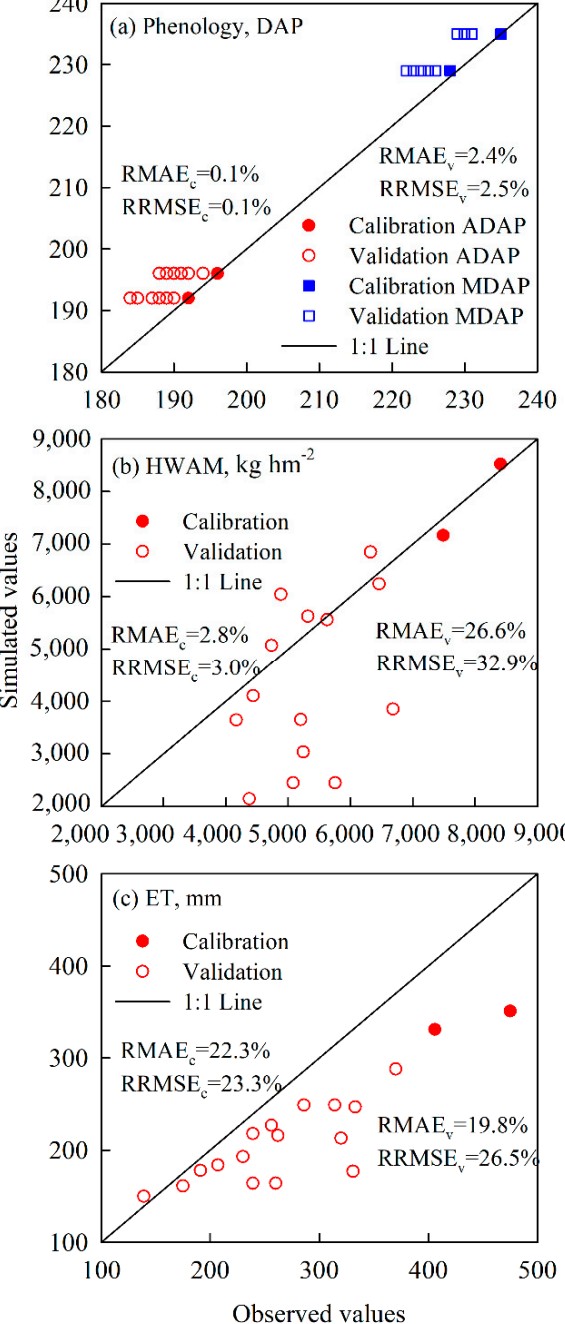

**Figure 2.** Calibration and verification performance of the CERES-Wheat model. The superscripts 'c' and 'v' show model calibration and validation processes, respectively. The abbreviations '*DAP*', '*ADAP*', '*MDAP*', '*HWAM*', and '*ET*' indicate the day after planting, anthesis date, maturity date, grain yields, and *ET*, respectively.

*3.2. Evaluation of Modified CERES-Wheat Model*

3.2.1. Performance of the Modified Crop Coefficient in the CERES-Wheat Model

Since the equations of $K_{CS}$ and water stress response function have been modified, the performance of using the modified $K_{CS}$ is then assessed stepwise in this section.

When $ET_0$ was calculated using the PM, the crop coefficient was a constant (1.0), while when computed using the PT, the crop coefficient changed to 1.1 with wheat growth, but in the early and late growing stages (Figure 3). However, the crop coefficient of wheat was different at different growth stages. With wheat's growth and development, the crop coefficient increased and afterward decreased. The changes in $KC_D$ and $KC_{Kang}$ align with the actual situation. In addition, the more severe the water stress, the more substantial the difference in crop coefficients estimated by different methods (Figure 3a,c). Before the jointing stage, there was no significant difference in crop coefficient between treatments. After jointing, the more severe the water stress, the lower the *LAI* of winter wheat, and the lower the leaf surface tendency to cover the ground, so the crop coefficient was smaller.

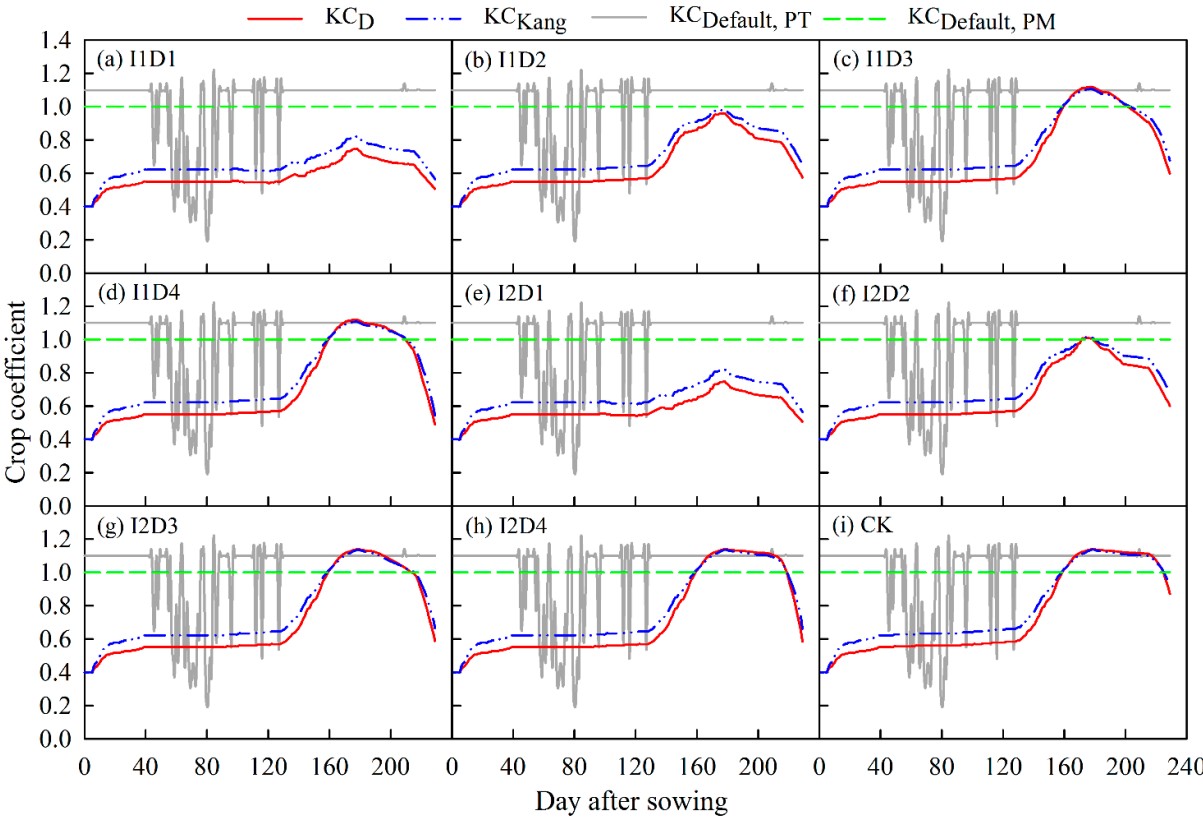

**Figure 3.** Dynamic changes in crop coefficients for winter wheat under different irrigation scenarios during the 2012–2013 seasons.

The water factor for winter wheat photosynthesis under different irrigation scenarios fluctuated with wheat growth during the 2012–2013 seasons (Figure 4). Our study showed that before model improvements, there were little differences in water factors between different treatments before jointing, which ranged from 0 to 0.48, indicating that winter wheat growth was subjected to water stress. However, the field experiment showed a minimal effect of water stress on wheat growth. Therefore, the model overestimated the impact of water stress on wheat growth. By improving the crop coefficients ($KC_D$ and $KC_{Kang}$), we observed that the water factors of different treatments were all 0, indicating that the early water stress had little effect on the growth of wheat, which was consistent with the results of the field experiments. However, severe water stress will limit the normal development of winter wheat and affect photosynthesis. Therefore, the water stress factor of 0 for the modified model still has some limitations, and further improvement is needed.

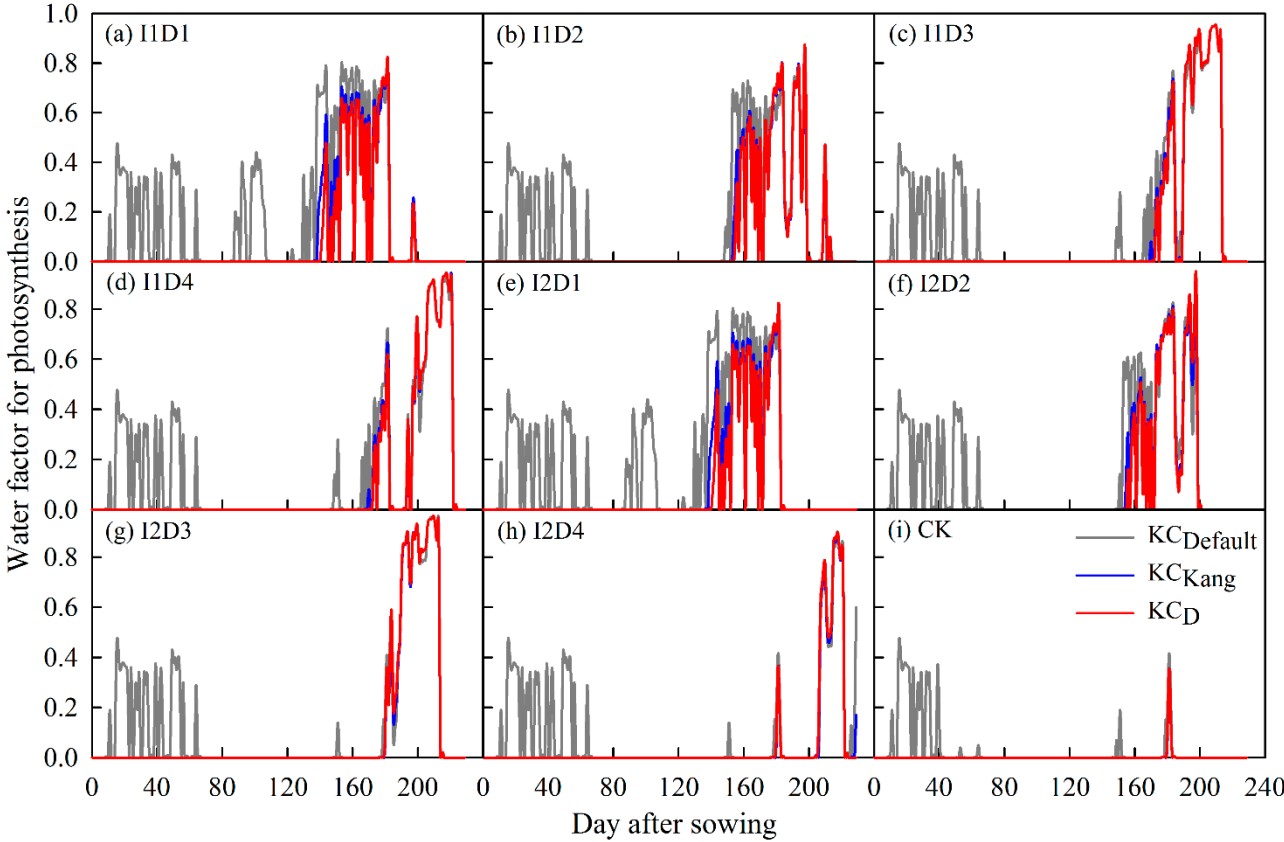

**Figure 4.** Dynamic changes in water factor for winter wheat photosynthesis under different irrigation scenarios during the 2012–2013 seasons.

3.2.2. Comparisons of Summary Output Variables Using Modified CERES-Wheat Model

Tables 3 and 4 show the performance of using the improved $K_{CS}$ in simulating *HWUM*, *CWAM*, and *HWAM* in the CERES-Wheat model. The original and modified model's simulated anthesis and maturity dates were the same values and, therefore, not considered in the analysis. The *RMAE* and *RRMSE* of *CWAM* and *HWAM* were lower when using the PM than the PT method in the original and modified model. However, the *RMAE* and *RRMSE* of *HWUM* were higher when using the PM than the PT method. In addition, the simulation accuracy of the modified model was more elevated than the original model (averaged *RMAE* and *RRMSE* were less than 30% in the modified model). In general, the highest simulation accuracy among the four scenarios is the scheme of PT_$KC_D$.

**Table 3.** Performances of static and dynamic crop coefficient in simulating *HWUM*, *CWAM*, and *HWAM*. The superscripts 'PT' and 'PM' indicate *ET* estimation methods in the CERES-Wheat model.

| Scheme | RMAE (%) | | | | RRMSE (%) | | | |
|---|---|---|---|---|---|---|---|---|
| | *HWUM* | *CWAM* | *HWAM* | *ET* | *HWUM* | *CWAM* | *HWAM* | *ET* |
| $KC_{Default}$ | 19.4 | 23.7 | 24.0 | 20.1 | 21.5 | 27.0 | 29.2 | 26.2 |
| PT $KC_{Kang}$ | 18.3 | 11.5 | 16.3 | 19.8 | 20.9 | 13.6 | 18.5 | 26.7 |
| PT $KC_D$ | 18.1 | 10.0 | 15.1 | 20.3 | 20.8 | 12.2 | 17.4 | 27.1 |
| PM $KC_{Kang}$ | 18.5 | 8.6 | 17.5 | 21.5 | 21.2 | 9.8 | 21.0 | 30.7 |
| PM $KC_D$ | 18.5 | 8.5 | 17.2 | 21.8 | 21.2 | 9.8 | 21.0 | 30.9 |

**Table 4.** Average performance of static and dynamic crop coefficients in simulated *HWUM*, *CWAM*, and *HWAM*.

| Scheme | RMAE (%) | RRMSE (%) |
|---|---|---|
| $KC_{Default}$ | 22.6 | 27.5 |
| $^{PT}KC_{Kang}$ | 16.5 | 19.9 |
| $^{PT}KC_D$ | 15.9 | 19.4 |
| $^{PM}KC_{Kang}$ | 16.5 | 20.7 |
| $^{PM}KC_D$ | 16.5 | 20.7 |

### 3.2.3. Comparisons of Time-Series Output Variables Using Modified CERES-Wheat Model

To adequately evaluate the modified CERES-Wheat model, we compared the simulation results of several crucial time-series output variables. The simulated dynamics of total soil water within depths 0–100 cm and winter wheat biomass varied greatly among treatments (Figures 5 and 6).

For total soil water, the dynamic simulation results showed that the simulation errors would become more prominent with the increase of soil water stress in the original model. The original model underestimated soil moisture at the early stage of winter wheat growth or overestimated the level of soil water stress, which probably contributed to the significant errors in biomass simulation (Figure 5a,b,e,f). However, the simulations and observations could match well when water stress occurred at the heading and grain-filling stages (Figure 5d,h). This confirmed that the CERES-Wheat model could correctly simulate soil water content when water stress occurred at later growth stages. After improving the CERES-Wheat model, the simulation accuracy improved, especially under low irrigation levels.

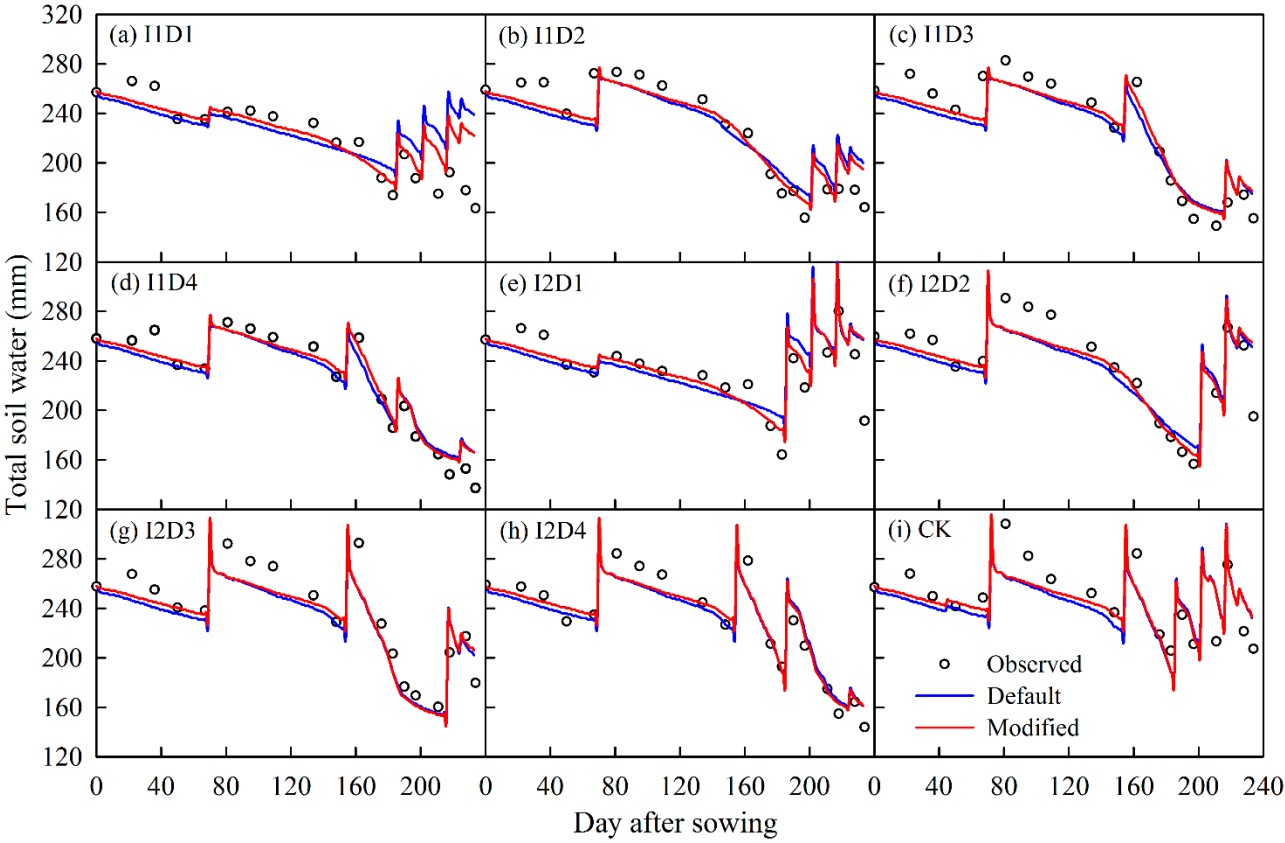

**Figure 5.** Dynamic changes in total soil water within depths 0–100 cm of winter wheat under different irrigation scenarios during the 2012–2013 seasons.

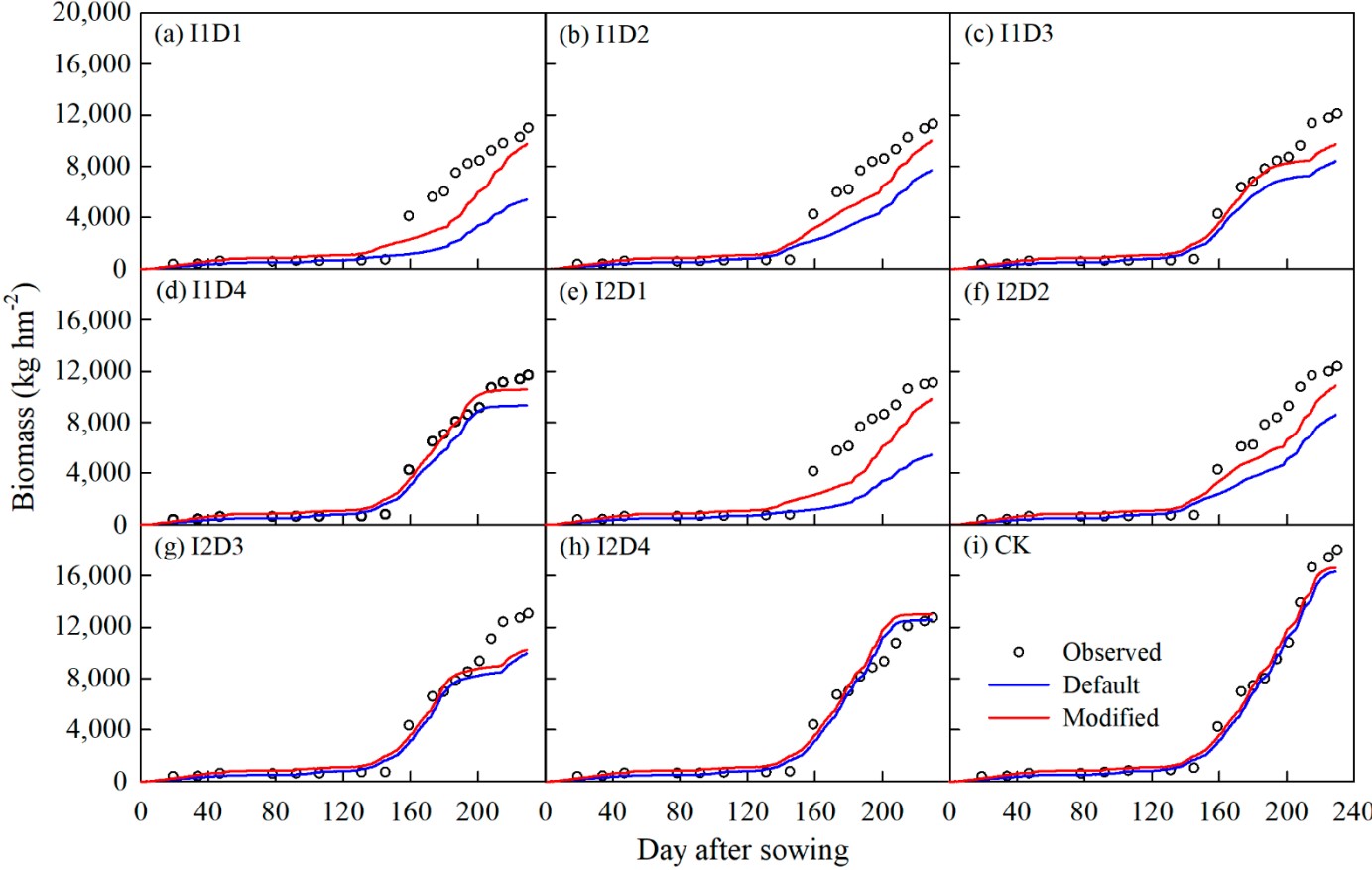

**Figure 6.** Dynamics of winter wheat biomass under different irrigation scenarios during the 2012–2013 seasons.

For biomass in winter wheat, either under low or high irrigation, if there were continuous water stress at early vegetative stages (e.g., wintering, returning green, and jointing stages), there would be significant errors in the simulation of biomass after the jointing stage since simulations were significantly lower than observations in the original model (Figure 6a,b,e,f). It overestimated the inhibition of winter wheat growth by soil water stress at returning green stage. However, the model simulated and observed results were very close when water stress occurred at the heading and grain-filling stages (Figure 6d,h). The modified model-simulated and observed results were very close when using the dynamic crop coefficient estimator simultaneously to improve the model (Figure 6b–d,f–h). It indicated that the model had been significantly improved. Despite improving the CERES-Wheat model, there were still some simulation errors when severe water stress occurred at wintering and returning green stages.

### 3.3. Validation of Modified CERES-Wheat Model

The validation results for winter wheat are presented in Table 5 for the two experiments in the DSSAT v4.7 databases. The modified (WSRF1 + $KC_S$ and WSRF1 + $KC_D$) and original *RMAE* results averaged across all measurements were nearly equal for the two cultivars. The *RRMSE* results were also roughly similar except for the output variable *HWAM*. The results indicated that water stress response function improved when simulating winter wheat growth under water stress conditions. However, the effect of crop coefficient improvement on the simulation results was uncertain. Most of the modified (WSRF1 + $KC_D$) *RMAE* and *RRMSE* results for cultivar 'NEWTON' were greater than the original results. In general, the improvement of the water stress response function did an excellent job of simulating winter wheat growth under drought conditions.

**Table 5.** Validation results of the CERES-Wheat model using different databases in the DSSAT v4.7. Superscripts '$^a$' and '$^b$' represent the validation cultivars 'NEWTON' and 'MARIS FUNDIN,' respectively.

| Items | RMAE (%) | | | | RRMSE (%) | | | |
|---|---|---|---|---|---|---|---|---|
| | *HWUM* | *CWAM* | *HWAM* | **Averaged** | *HWUM* | *CWAM* | *HWAM* | **Averaged** |
| $^a$ Default | 21.0 | 22.4 | 19.3 | 20.9 | 24.4 | 21.6 | 21.5 | 22.5 |
| $^a$ $KC_{Kang}$ | 23.9 | 21.0 | 16.0 | 20.3 | 28.2 | 24.6 | 26.2 | 26.3 |
| $^a$ $KC_D$ | 25.2 | 21.2 | 16.7 | 21.1 | 29.2 | 25.0 | 26.7 | 27.0 |
| $^b$ Default | 11.3 | 16.4 | 11.1 | 12.9 | 14.0 | 18.0 | 11.2 | 14.4 |
| $^b$ $KC_{Kang}$ | 9.9 | 13.6 | 11.7 | 11.7 | 12.2 | 17.1 | 11.9 | 13.7 |
| $^b$ $KC_D$ | 9.3 | 13.7 | 11.8 | 11.6 | 11.9 | 17.1 | 12.0 | 13.7 |

## 4. Discussions

The CERES-Wheat model was able to correctly simulate the growth and yield of winter wheat under sufficient water conditions. The simulation accuracy of winter wheat yield was relatively high, which agreed with some previous studies [16,51,57]. However, the simulation accuracy of the CERES-Wheat model became low when the water deficit occurred at the early vegetative stage. It usually overestimated unit grain weight but underestimated final *ET* and grain yield [58]. There are different reasons for large simulation errors. Nouna et al. [59] found the model did not sufficiently simulate *LAI* during the whole season. Yao et al. [60] suggested permanent wilting point plays an important role in simulating winter wheat growth under water deficit conditions. These soil parameters should be measured more directly and accurately and need to be further tuned based on field observations when simulating crop growth under serious water stress conditions. Dejonge et al. [39] found the evapotranspiration was overestimated or underestimated when water stress occurred at the different growth stage. As a result, they provided a new equation that calculates a dynamic crop coefficient as a function of *LAI* to improve the accuracy of simulated *ET* [39].

In this study, the ET was estimated by PT (PM) method and the improved *Kc*s. The simulation accuracy of the improved model was higher than that of the original model, especially under low irrigation levels. The simulation results of *HWUM*, *CWAM*, and *HWAM* are significantly improved, but the simulation results of *ET* are not improved. The simulation errors may be first attributed to the low accuracy of water stress. Because the effect of water stress on crop growth is very complex, the current linear function can not quantify the effect of water stress on the crop growth process. Our results show that the model underestimates ET, which is consistent with the results of Kang et al. [61]. The simulation accuracy of *ET* is not high, which may be due to the large variation in seasonal precipitation or the dependence of the calculation of potential *ET* on simulated *LAI* [41,61]. Although the *ET* simulation accuracy of the improved model is not improved, *HWUM*, *CWAM*, and *HWAM* are significantly improved, which is the same as the results of previous studies. The increase in *HWUM*, *CWAM*, and *HWAM* may be attributed to the lower irrigation level, which led to the downward growth of wheat roots and the absorption of more nutrients.

In general, the CERES-Wheat model has some drawbacks in simulating winter wheat growth under complicated scenarios of soil water stress. For better applications to winter wheat management in arid and semi-arid regions, the existing CERES-Wheat model needs to be further improved. The process of crop growth is complex nonlinear; a simple linear relation to describe the water stress response function may not accurately describe the crop response process. In future studies, not only the influence of soil parameters and crop coefficients on the CERES-Wheat model but also the influence of nonlinear changes in water stress on the simulated results should be considered.

## 5. Conclusions

In this study, a dynamic crop coefficient was used to improve the simulation accuracy of the CERES-Wheat model under water stress. The error of the PM-based method is larger, while the accuracy of the PT-based method is higher. Among the four scenarios, PT_$KC_D$ has the better simulation accuracy, and the simulation accuracy is improved the most. Although the simulation effect of the model on *ET* is not much improved, the simulation accuracy of *HWUM*, *CWAM*, and *HWAM* is significantly improved. Therefore, the improved CERES-Wheat model can better simulate winter wheat growth in arid and semi-arid regions under water stress conditions. Under the future climate change scenario, it is very important to determine the optimal irrigation amount to achieve a high yield. To be effectively applied in winter wheat management in arid and semi-arid regions of China, we suggest that the existing CERES-Wheat model should be further verified, modified, and improved by fully considering the influence of various factors, such as the nonlinear water stress index.

**Supplementary Materials:** The following are available online at https://www.mdpi.com/article/10.3390/agriculture12111902/s1, Table S1: Soil parameters of the three experimental field sites.

**Author Contributions:** Data curation, X.L. and Z.H.; funding acquisition, T.J.; methodology, O.O.A.; project administration, N.Y.; writing—original draft, Y.W. and H.R. All authors have read and agreed to the published version of the manuscript.

**Funding:** This research was partly supported by the Natural Science Foundations of China (U2243235, 52209070, and 52079114), the Foreign Young Talent Plan (No. QN2022172005L), the Chinese Universities Scientific Fund (2452020211), and the High-end Foreign Experts Introduction Project (No. G2022172025L).

**Institutional Review Board Statement:** Not applicable.

**Data Availability Statement:** The data presented in this study are available on-demand from the first author at (weiyingnan1998@163.com).

**Conflicts of Interest:** The authors declare no conflict of interest.

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
