# Peer review of "Better Performance of the Modified CERES-Wheat Model in Simulating Evapotranspiration and Wheat Growth under Water Stress Conditions"

_agriculture, doi:10.3390/agriculture12111902_

Round 1

Reviewer 1 Report

I have gone through the manuscript. A very informative topic and scientific conclusion on Evaluating the performance of the modified CERES-Wheat 2 model in simulating evapotranspiration and wheat growth un- 3 der water stress conditions . Overall, this manuscript is well written.   I have addressed several major comments as follows:

-         The introduction should be supported with sufficient information about the CERES-Wheat 2 model with relevant references.  Also, Introduction part must contains the whole background regarding the targeted problem and how to solve that problem with comparison with literature review; please check and revised accordingly.

-         The novelty of the study needs to be highlighted compared to other similar studies.

-         More explanation should be given about CERES-Wheat 2 model . It is better to mention it in detail

-         What is the Wheat variety?

-         It is very cool to focus on the CERES-Wheat 2 model, but some indications of biochemical stress must be studied What type of cultivated soil?

-         More details should be mentioned about the dates of sampling and in what part of the plant they were taken?

-         Materials and methods section must contain recent and related references with more details to be beneficial to broad scientific readers.

-         Discussion needs enhancement with real explanations not only agreements and disagreements.

-         Conclusion part must contain the importance of paper, the future work and novelty.

Reviewer 2 Report

Comments on agriculture-1961473

This manuscript presents a potentially interesting and significant question, that is improve the simulation accuracy of the CERES-Wheat model under water stress in arid and semi-arid regions using the winter wheat experiment data from 2012 to 2014. The authors provided a reliable reference of about dynamic crop coefficients using experiment data from different sites and total soil water dynamics accuracy and winter wheat yield-related indices simulated
by the non-modified and modified CERES-Wheat model. The authors fail to present the importance of the study depending on available data. Also, necessary information in materials and methods are absent i.e. the conditions of wheat growth in different sites. However, Discussion needs more and more amendments and, have to be more stand on the CERES-Wheat model to simulate Wheat growth in arid and semi-arid regions (data are from China, USA and UK) under water stress conditions. Also, the discussion needs more explanation’s for significant results.

Line 34: Keywords: add new keywords and don’t repeated title words

Line 36: Introduction don’t provide sufficient background and must be included recent and relevant references. The lack of recent references and an inadequate presentation of the results significantly altered the interest and scope of this work.

L 376: References:

DOI of all references must be add

Journal name must be italic

All scientific names are italic (e.g., lines, 420)

Author Response

Please read the attachment.

Reviewer 3 Report

The manuscript contains certain number of mistakes that could be revised. This manuscript is useful research on topic of high interest and experimentally consistent

The discussion is very long and not particularly informative. The introduction and the discussion should be much more specific and quantitative.

Most of the discussion sounds, even if in some steps it looks more speculative than based on well-proven and explained results. At present the logic of the study is hard to follow.

 Lastly, this is a long paper with many tables and supplemental tables and figures. The figure and table’s numbers are not in the right order

In order to give your work a fair chance of being accepted, we have to make every effort possible to secure reader understanding.   If we don’t get the English right, the value of the work diminishes
Even if the required most important minor revision will take time, the Authors should integrate missing points and improve the weakness of the work, before to resubmit the paper.

Author Response

Please read the attachment.

Round 2

Reviewer 1 Report

 Accept in present form

Reviewer 2 Report

no more comments